# TwIdw—A Novel Method for Feature Extraction from Unstructured Texts

**Kitti Szabó Nagy [1,\*] and Jozef Kapusta [1,2]**

[1] Department of Informatics, Faculty of Natural Sciences and Informatics, Constantine the Philosopher University in Nitra, 949 01 Nitra, Slovakia; jkapusta@ukf.sk

[2] Institute of Computer Science, Pedagogical University of Cracow, 30-084 Kraków, Poland

[\*] Correspondence: kitti.szabo.nagy@ukf.sk

**Featured Application: The research has a potential application in the field of fake news detection. By using the feature extraction technique, TwIdw, proposed in this paper, more relevant and informative features can be extracted from the text data, which can lead to an enhancement in the accuracy of the classification models employed in these tasks.**

**Abstract:** This research proposes a novel technique for fake news classification using natural language processing (NLP) methods. The proposed technique, TwIdw (Term weight–inverse document weight), is used for feature extraction and is based on TfIdf, with the term frequencies replaced by the depth of the words in documents. The effectiveness of the TwIdw technique is compared to another feature extraction method—basic TfIdf. Classification models were created using the random forest and feedforward neural networks, and within those, three different datasets were used. The feedforward neural network method with the KaiDMML dataset showed an increase in accuracy of up to 3.9%. The random forest method with TwIdw was not as successful as the neural network method and only showed an increase in accuracy with the KaiDMML dataset (1%). The feedforward neural network, on the other hand, showed an increase in accuracy with the TwIdw technique for all datasets. Precision and recall measures also confirmed good results, particularly for the neural network method. The TwIdw technique has the potential to be used in various NLP applications, including fake news classification and other NLP classification problems.

**Keywords:** feature extraction; dependency grammar; modified TfIdf; neural networks

## 1. Introduction

The detection and classification of fake news have been topics of interest in the field of Natural Language Processing (NLP) for several years. Various techniques have been utilized to address these issues, including supervised and unsupervised learning, deep learning, and feature engineering [1].

This work focuses on the NLP field, in which a suitable dataset with unstructured text is a crucial requirement for verifying research results. For this reason, the primary focus was placed on the area of fake news detection. There is no clear definition for fake news, but some researchers specify fake news as deceptions, gossips, frauds, misinformation or hoaxes, emphasizing their significant impact on society [2,3]. More commonly, fake news refers to news that can be debunked and substantiated with evidence. Humanity is compelled to seek new types of defense mechanisms against fake news. Many studies have focused on developing techniques to identify and classify fake news on social media platforms. Overall, the detection and classification of fake news in NLP is an active research area, with many techniques and approaches being proposed and studied [1,4].

Machine learning algorithms have been commonly used to classify news articles. support vector machines (SVM), random forest, and deep learning techniques are the most

common algorithms used in the studies [5–7]. Feature engineering is another widely used approach that extracts various features from news articles to classify them as either real or fake. These techniques have shown promising results, with several studies achieving high accuracy in classifying fake news articles [8–10].

In this research, the focus was placed on the identification of fake news based on their contents. When processing natural language, it is necessary to define the levels of language that will be used for classification.

The emphasis of this study lies in examination at the syntactic level, encompassing the relationships between words within a sentence. Syntactic analysis is fundamental for analyzing language at higher levels such as semantic, morphemic, or contextual. Syntax is concerned with identifying sentence elements and their relationships [11]. In NLP, it is also known as dependency grammar or dependent syntax [12]. The outcome of dependency analysis is the dependency structure of the sentence, which is often represented in the form of a tree [13].

In this research, a novel NLP technique is presented, TwIdw, for creating word embeddings that preserve the importance of words in a sentence. TwIdw is based on the concept of dependency trees and utilizes information about the structure of a sentence and the frequency of word occurrence in a corpus to assign a unique weight to each word. UDPipe tool is used for the dependency analysis [14]. The effectiveness of the TwIdw technique is evaluated by conducting a comparison with the commonly used NLP technique as TfIdf. The study aims to investigate whether the extracted knowledge from text using dependency grammar and its subsequent use for word embedding leads to higher accuracy in classifying the fake news.

The research study is organized as follows. The review of previous NLP research and the commonly used methods for identifying fake news are introduced in Section 2—Related Work. The study builds upon these methods by incorporating dependency grammar to extract knowledge from unstructured texts and their usage for word embedding. In the subsequent Section 3—Materials and Methods—the methodology, datasets, and tools used in the analysis are described. Finally, the results are presented in Section 4—Results—in which the implications are also discussed and suggestions for future research in this area are summarized. Section 6—Conclusions—describes the impact of the research and offers thoughts on future opportunities within this area of study.

## 2. Related Works

An overview of existing studies and research conducted in the field of NLP is here provided, with a focus on highlighting the key findings and limitations of previous works. Sarcasm detection, sentiment analysis, fake news detection, and text summarization are all prominent tasks within the field of NLP. Ongoing research aims to improve the performance and robustness of these NLP tasks to meet the ever-growing demands of language understanding, classification, and text processing in real-world applications.

Singh and Srivastava [15] created a machine learning model which detects sarcasm. They used Twitter tweets as dataset. The goal was to enhance the performance of existing sarcasm detection algorithms for Hindi–English language combinations. This was achieved by adding a data balancing layer in the algorithm before classifying the extracted features. They achieved an f1 score 0.97.

Secondly, sentiment analysis aims to determine the underlying sentiment or emotion expressed in the texts, ranging from positive to negative or neutral. This task is essential for understanding public opinion, customer feedback, and social media sentiment. Shahi et al. [16], in their study, analyzed people's sentiment on COVID-19 in the Nepali language using both syntactical and semantic information from social media posts. Two text representation methods (TfIdf and FastText) are combined to achieve hybrid features and nine machine learning classifiers are implemented. Results show that the hybrid feature extraction method outperforms other methods and provides excellent performance compared to state-of-the-art methods on a publicly available Nepali COVID-19 tweets dataset.

Twitter is attracting researchers who utilize it for sentiment extraction and data mining, as it has emerged as the preferred social media platform for users to express their opinions and thoughts about businesses and services. Mishra et al. [17] employ knowledge graphs and K-means clustering to develop an innovative approach for sentiment analysis, highlighting the connection between the tourism sector and the economy. Clustering techniques were applied to group similar entities, forming a cluster-wise knowledge graph that visualizes relationships among various factors connecting tourism and the economy.

In another study, Neogi et al. [18] collected around 20,000 tweets to analyze the sentiment. Bag of words and TfIdf were used as models. The analysis revealed that Bag of words outperformed TF-IDF in sentiment classification. Naive Bayes, decision trees, random forests, and support vector machines were employed, with random forest achieving the highest classification accuracy.

Syntactic analysis, also known as parsing, plays a crucial role in NLP by analyzing the grammatical structure of sentences. The importance of syntactic analysis lies in its ability to understand sentence structure. By incorporating syntactic analysis into NLP systems, the accuracy, interpretability, and overall performance of various language processing tasks are enhanced. Nagy and Kapusta [19] pre-processed a dataset of fake and real news using syntactic analysis and used dependency grammar to determine the importance of words within sentences. They created input vectors for classification as TfIdf weighted by dependency grammar depths and compared it with the TfIdf method. Their results show that using dependency grammar information can improve the classification of fake news with acceptable accuracy and even improve existing techniques such as TfIdf.

Morphological analysis involves analyzing the internal structure and forms of words in a language. Kapusta et al. [20] compared fake and real news based on morphological tags. They found statistically significant differences in verbs and nouns between the real and fake news. Kapusta et al. [21] experimentally evaluated the potential of n-grams and POS tags for the classification of fake and true news related to the COVID-19 pandemic. Three techniques based on POS tags were proposed and applied. The performance measures of the proposed techniques were compared with the standard TfIdf technique. The results showed that the proposed techniques are comparable with TfIdf, and morphological analysis can improve the baseline TfIdf technique. The study suggests that the precision for fake news and recall for real news can be statistically significantly improved.

In NLP classification tasks, machine learning and deep learning techniques have been extensively employed. Machine learning algorithms, such as support vector machines (SVM), decision trees, and random forests, have been utilized to build models that can classify text into predefined categories. In recent years, deep learning models have emerged as powerful tools for NLP classification tasks. Deep neural networks, including convolutional neural networks (CNNs), recurrent neural networks (RNNs), and transformer models, have shown remarkable performance and allowing for more accurate and robust classification. Haque et al. [22] compare multiple machine learning and deep learning models to identify suicidal thoughts on Twitter with the goal of achieving high accuracy to recognize early indications and prevent suicide attempts. The study used text pre-processing and feature extraction techniques, trained several models, and experimented on a dataset of 49,178 instances. The findings show that the random forest model achieves the highest classification score among machine learning algorithms with an accuracy of 93% and f1-score of 0.92. However, training deep learning classifiers with word embedding increases the performance of ML models, with the BiLSTM model achieving an accuracy of 93.6% and f1-score of 0.93. This means that the BiLSTM model is performing better than the random forest model.

Madani et al. [23] proposed a model for detecting fake news using feature extraction and machine learning algorithms. The sorted news samples are sent to long short-term memory and classical machine learning algorithms for detection. The model performs better than benchmark models in detecting fake news. In the FakeNewsNet database, the proposed model had an AUC of 69%, and in the Liar database, the proposed model had

an accuracy of 70%. Mehta and Mishra [24] focus on utilizing various machine learning models to detect fake news in COVID-19-related tweets. Gradient Boosting Classifier, Logistic Regression, random forest Classifier, and Decision Tree Classification models were employed to classify tweets as either "Fake News" or "Not Fake News" in relation to COVID-19. Among these models, Logistic Regression emerged as the best performer, exhibiting the highest f1-score of 93%.

In conclusion, one of the most crucial issues in NLP research, due to its significant impact on society, is fake news detection. Various techniques have been proposed to address this problem, including classical machine learning methods and deep learning models. These methods typically rely on feature extraction and representation techniques. The methods in the analyzed literature have shown promising results, with some models achieving high accuracy and AUC scores in detecting fake news in various datasets. However, there is still room for improvement, and future research should focus on developing more effective feature extraction techniques and exploring new deep learning architectures to further enhance the performance of these models. Feature extraction plays a vital role in machine learning models, enabling them to understand and analyze unstructured texts. Developing new and innovative feature extraction techniques can lead to improved accuracy in various NLP tasks, such as sentiment analysis, text classification, named entity recognition, and fake news detection. By capturing more relevant and informative features from unstructured texts, models can make more accurate predictions and classifications.

## 3. Materials and Methods

This study aims to investigate the impact of a novel feature extraction technique, TwIdw (Term weight–Inverse document weight), on the accuracy of text classification models. The methodology followed for the study is visualized in Figure 1 and is described below.

a.  Data Collection and Pre-processing: Different datasets are used in the research. Data are described in Section 3.1. The data underwent pre-processing to eliminate any irrelevant information and duplicates;

b.  Feature Extraction: Two feature extraction techniques are used: TfIdf and TwIdw. For TfIdf, the frequency of occurrence of each term is calculated; and for TwIdw, the depth of each term is used. UDPipe is used to calculate the depth of each term;

c.  Model Training: Text classification models random forest and feedforward neural networks are trained using both TfIdf and TwIdw features. The models are evaluated using evaluation metrics: accuracy, precision, recall, and F1 score;

d.  Results Analysis: The results obtained from the models are analyzed to determine the impact of TwIdw on the accuracy of text classification. The results are compared with those obtained using TfIdf. The F1 score is statistically evaluated;

e.  Conclusion: The findings of the study are discussed, and conclusions are drawn regarding the effectiveness of TwIdw in improving the accuracy of text classification models. Limitations of the study are also highlighted, and suggestions for future research are made.

### 3.1. Datasets

In this work, three datasets containing true and false news about the COVID-19 disease were utilized. The first dataset, freely available [25], had 1164 records, out of which 1154 were usable after cleaning. The second dataset, consisting of manually annotated data related to COVID-19 disease [26], comprised 2720 usable records and exhibited an unbalanced distribution. The third dataset, named KaiDMML and freely available [27], encompassed true and false news from political life. Originally created for the FakeNewsTracker tool, this dataset underwent manual annotation and contains 407 records.

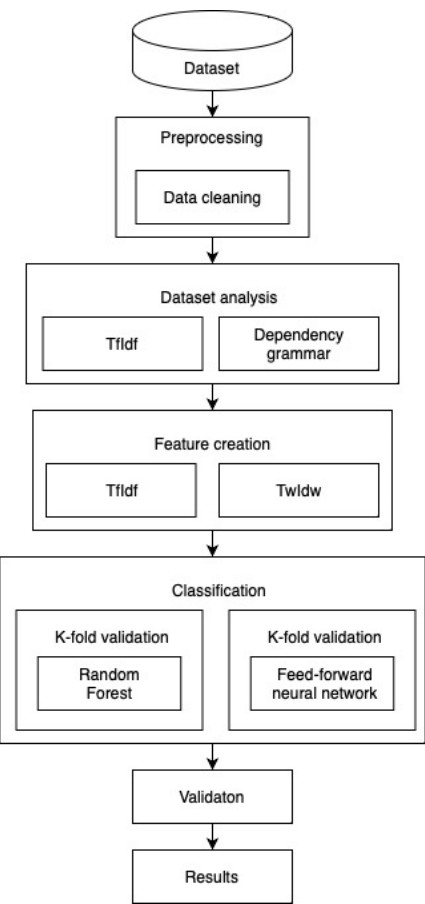

**Figure 1.** Research methodology.

### 3.2. Feature Extraction

Two feature extraction techniques are used: TfIdf and the proposed TwIdw. It is believed that TwIdw will carry greater informative value as it is based not only on word frequency but also on dependency tree, while TfIdf only captures the frequency of individual words. Due to the higher informative character of the values obtained using the TwIdw technique, an improvement in the classification accuracy is expected when it is employed in the classification model. The tool UDPipe is used to calculate the dependency depth of each term.

TfIdf is a base technique for feature extraction. Its calculation is as Equations (1)–(3) shows, where t is a term, d is a document, w is the frequency of the terms in the document, D is a document corpus, and N is the number of documents in the corpora.

$$\text{Tf}(t, d) = \frac{f(t, d)}{f(w, d)} \tag{1}$$

$$\text{Idf}(t, D) = \ln \frac{N}{\sum (d \in D : t \in d) + 1} \tag{2}$$

$$\text{TfIdf}(t, d, D) = tf(t, \ d) \times idf(t, D) \tag{3}$$

TfIdf measures the importance of a term in a document or corpus of documents by considering not only the frequency of the term in the document but also the frequency of the term in the entire corpus. The Tf component measures how frequently a term occurs in a document, while the Idf component measures how rare or common the term is across all documents in the corpus. By combining these two components, the TfIdf score can effectively highlight terms that are important and relevant to a particular document or

topic, while filtering out terms that are common across all documents. TfIdf has been widely used in various NLP tasks such as text classification, information retrieval, and topic modeling, and has been shown to be effective in improving the performance of these tasks.

Dependency trees, also known as dependency parse trees, are used in linguistics and NLP to represent the grammatical relationships between words in a sentence. While syntactic trees focus on phrase structure, dependency trees focus on the relationships between individual words. UDPipe [14] is a NLP tool that provides tokenization, part-of-speech tagging, morphological analysis, and dependency parsing for text data in various languages. It is designed to be easy to use and integrate into existing workflows, making it a popular choice for many NLP projects. The output generated by UDPipe can be used in various applications, such as machine translation, named entity recognition, sentiment analysis, and text classification.

The UDPipe tool operates by taking raw text input and breaking it down into individual words or tokens. Each token is then assigned a part-of-speech tag (such as noun, verb, adjective, etc.). The grammatical relationships between the tokens are identified in the form of a dependency tree. A dependency tree, which is generated by UDPipe, represents the grammatical structure of a sentence graphically. The nodes in the tree represent words in the sentence, and the edges represent the relationships between them. The part-of-speech tag is assigned to each word in the tree, and the tree is typically rooted at the main verb of the sentence. The result of UDPipe dependency parsing is presented in Table 1, and it is visualized in Figure 2. In the example, the root word of the sentence is "labeled", and it has a depth of 0. The words "word", "is", and "part" are dependent on the root word and have a depth of 1.

**Table 1.** Result from dependency parsing with calculated depth.

| Id | Form | Lemma | UPosTag | Head | Calculated Depth |
|----|------|-------|---------|------|------------------|
| 1 | Each | each | DET | 2 | 2 |
| 2 | word | word | NOUN | 7 | 1 |
| 3 | in | in | ADP | 5 | 3 |
| 4 | the | the | DET | 5 | 3 |
| 5 | tree | tree | NOUN | 2 | 2 |
| 6 | is | be | AUX | 7 | 1 |
| 7 | labeled | label | VERB | 0 | 0 |
| 8 | with | with | ADP | 10 | 2 |
| 9 | its | its | PRON | 10 | 2 |
| 10 | part | part | NOUN | 7 | 1 |
| 11 | of | of | ADP | 13 | 3 |
| 12 | speech | speech | NOUN | 13 | 3 |
| 13 | tag | tag | NOUN | 10 | 2 |
| 14 | . | | PUNCT | 7 | 1 |

The proposed TwIdw is set to Term weight–inverse document weight. The intuition behind using term depth is that words that are closer to the root of the sentence tree are more important and carry more meaning than words that are farther away. The calculation of this technique is based on TfIdf, but the term frequencies are replaced by the depth of the words in documents. Term depth refers to the distance of a word from the root of the sentence tree in a dependency parse, which is calculated using the UDPipe tool [14].

$$Tw(t, d) = \frac{\frac{\sum w(t,d)}{N(t,d)}}{N(w, d)} \tag{4}$$

$$Idw(t, D) = \ln\frac{max(N(w, D))}{max(w(t, D))} + 1 \tag{5}$$

$$TwIdw(t, d) = Tw(t, \ d) \times Idw(t, D), \tag{6}$$

where t is a term, d is a document, w is an inverse of depth value, D is a text corpus, and N is a quantity. The proposed approach for creating vectors ensures that the significance of words is preserved. This is achieved by assigning higher TwIdw values to words closer to the root of the dependency tree, indicating their greater importance in the sentence structure. Furthermore, the approach considers the frequency of word occurrence in the corpus, which is inversely proportional to its TwIdw value. In other words, words that occur more frequently in the corpus are assigned lower TwIdw values.

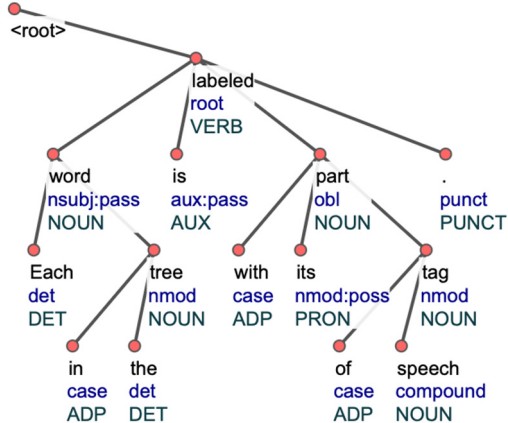

**Figure 2.** Dependency tree generated by UDPipe.

*3.3. Models*

In the study, two models for the classification task were explored: a random forest model and a feedforward neural network. The random forest model is an ensemble learning method that constructs a multitude of decision trees during training and outputs the class that is the mode of the classes [28]. A total of 100 trees were utilized in the forest, and Gini impurity was employed as the function to measure the quality of a split. No maximum depth or minimum samples were specified in the model. On the other hand, the feedforward neural network is a type of artificial neural network that uses a set of input, hidden, and output layers to model complex relationships between inputs and outputs [29]. The network architecture is as follows: 4 hidden layers, with 1000, 500, 60 and 8 neurons and sigmoid activation function. Both models were trained on the labeled dataset and subsequently evaluated using various performance metrics, which will be presented in the results section.

According to the methodology described in Section 3, vectors were created based on the proposed technique TwIdw, as well as the reference technique TfIdf. Models for identifying fake news were created using the random forest and feedforward neural network methods.

K-fold cross-validation was used for evaluating the created models. K-Folds technique is popular and easy to understand, it generally results in a less biased model compared to other methods and it ensures that every observation from the original dataset has the chance of appearing in training and test set. The advantage of this method is that all observations are used for both training and validation, and each observation is used for validation exactly once. When evaluating all the created models, a value of k = 10 was utilized. Specifically, the performance of these models in terms of accuracy, precision, recall, and f1-score are reported.

## 4. Results

The achieved results for accuracy (Table 2) are not decisively in favor of the proposed TwIdw technique, but in most cases, there was an increase in the accuracy of fake news classification. However, in most cases, the technique resulted in better accuracy for fake news classification. There were only two cases where the accuracy was lower than other methods, both when classifying the COVID dataset using the random forest algorithm. In these cases, the TwIdw technique resulted in 0.2% and 0.7% lower accuracy, respectively. In all other cases, the TwIdw technique achieved higher accuracy.

**Table 2.** Accuracy of models.

| Technique/Method | Random Forest | Neural Network | Random Forest | Neural Network | Random Forest | Neural Network |
|---|---|---|---|---|---|---|
| | COVID-19 Auto | | COVID-19 Manual | | KaiDMML | |
| TfIdf | 90.2% | 93.6% | 83.6% | 84.9% | 75.3% | 77.8% |
| TwIdw | 90.0% | 94.2% | 82.9% | 85.3% | 76.3% | 81.7% |

The highest accuracy for classifying fake news (94.2%) was achieved by the TwIdw technique combined with a neural network on the automatically evaluated COVID-19 dataset. A marked improvement in accuracy was observed when classifying the political dataset using a neural network, where the TwIdw technique resulted in a 3.9% increase in accuracy.

In terms of the precision metric, there were no significant increases in values observed. The achieved minimum precision values were higher in all cases of k-fold validation except for the classification of the political dataset using a neural network (Figures 3–5). In the case of the COVID-19 automatic dataset combined with a neural network, the TwIdw technique had the smallest interquartile and variance range. Overall, the best results for precision were achieved using the TwIdw technique on the automatically evaluated COVID-19 dataset. The recall metric also confirms the change of interquartile and variance range (Figures 6–8).

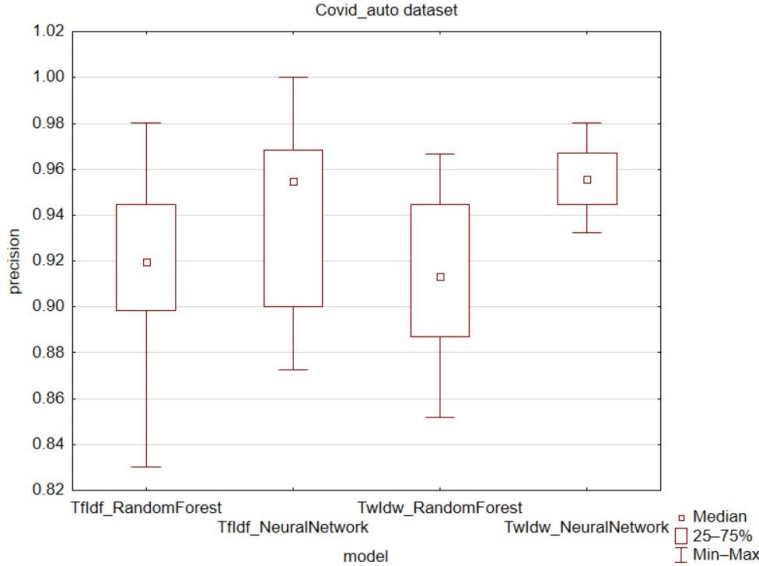

**Figure 3.** Precision metric for the COVID_auto dataset.

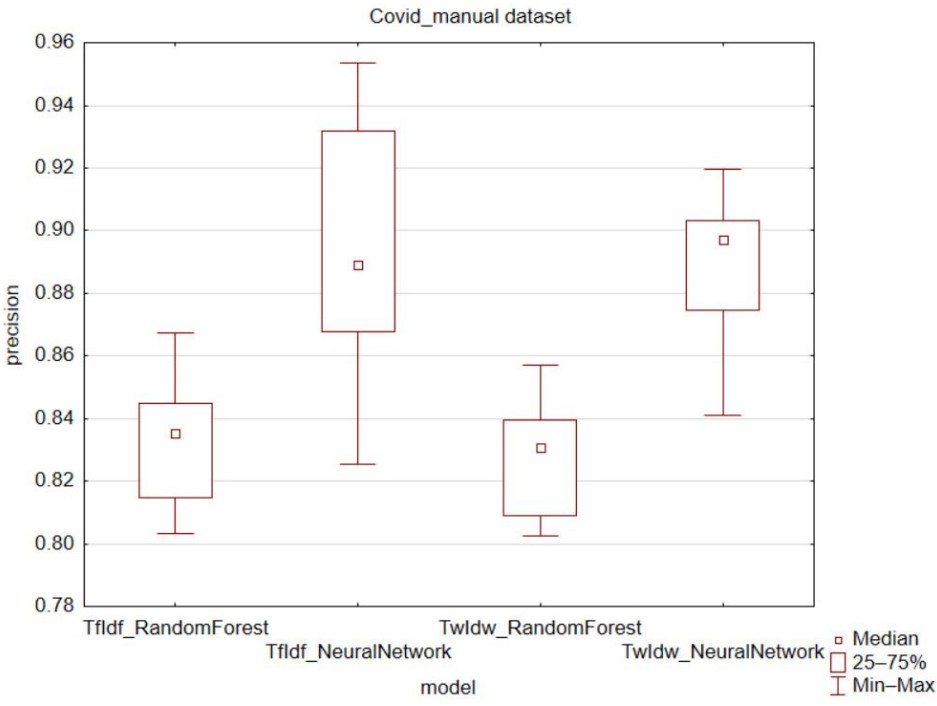

**Figure 4.** Precision metric for the COVID_manual dataset.

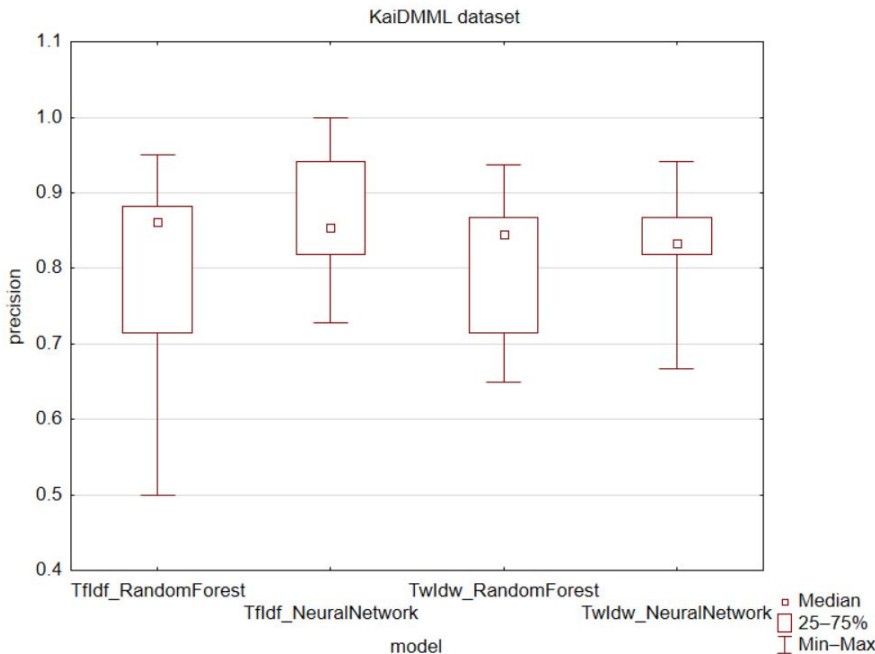

**Figure 5.** Precision metric for the KaiDMML dataset.

In terms of performance measures, better results were observed in favor of the TwIdw technique, especially for the feedforward neural network model. Recall indicates the rate of correctly identified fake news and real news. It showed better results for the TwIdw technique. In the case of the COVID_auto dataset (Figure 6), the results were inconclusive, especially for the random forest model, where the TwIdw technique showed an increase in the minimum observed value and a decrease in the quartile and variation range. However, the median, upper quartile, and lower quartile values were lower. For the COVID_auto dataset, the feedforward neural network model showed higher median, maximum, and minimum values, as well as higher upper quartile values in favor of the TwIdw technique.

For the COVID_manual dataset (Figure 7) and the recall metric, better results were observed for the traditional TfIdf technique in the case of the random forest model. The results for the feedforward neural network model were mixed, with only higher median and lower quartile values observed. However, in both models, the variation range and quartile range decreased for the recall metric.

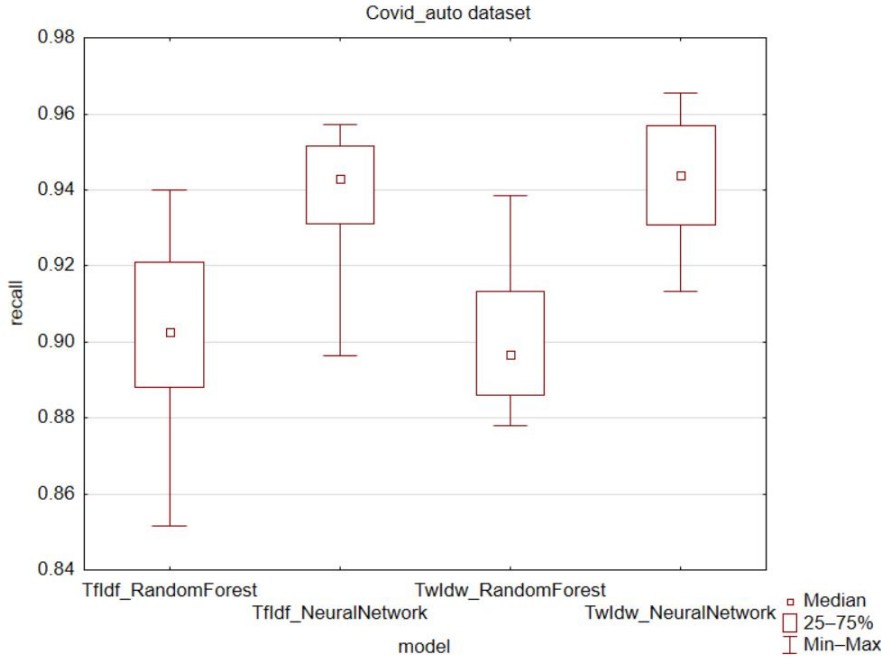

**Figure 6.** Recall metric for the COVID_auto dataset.

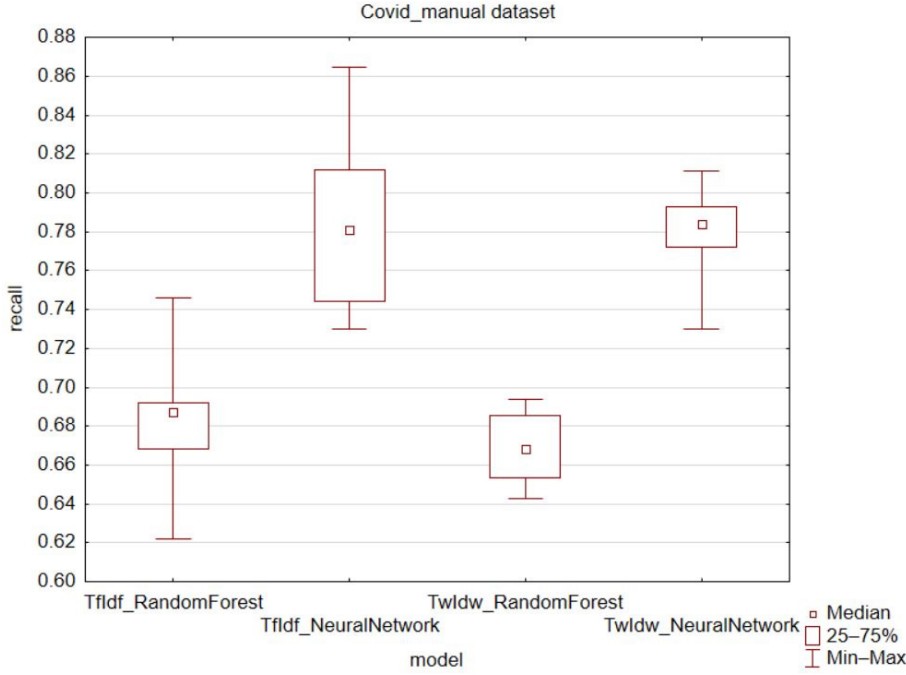

**Figure 7.** Recall metric for the COVID_manual dataset.

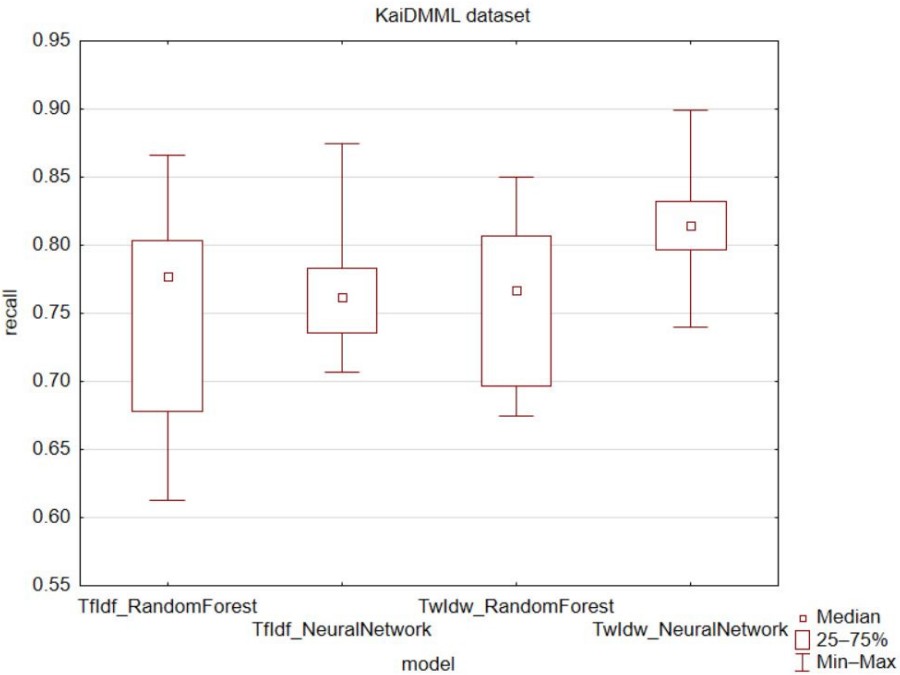

**Figure 8.** Recall metric for KaiDMML dataset.

For the KaiDMML dataset (Figure 7) and the recall metric, the results were ambiguous for the random forest model, like the previous datasets. However, for the feedforward neural network model, all values were higher (max, min, median, upper quartile, lower quartile). For both models, there was a decrease in the variance and quartile range.

The f1-score metric combines both precision and recall metrics, giving them equal weight. This metric confirmed an increase in values for the proposed TwIdw technique (Figures 9–11). Worse results for the f1-score metric were only observed in the manually created COVID-19 dataset when using the random forest classification algorithm. In all other cases, there was an increase in the average value of the f1-score metric.

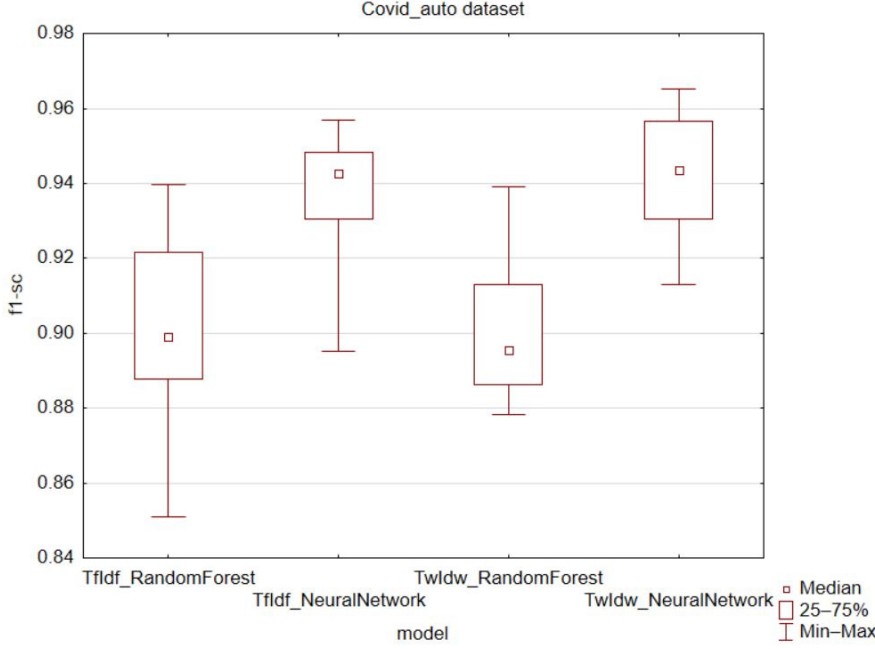

**Figure 9.** F1 score metric for the COVID_auto dataset.

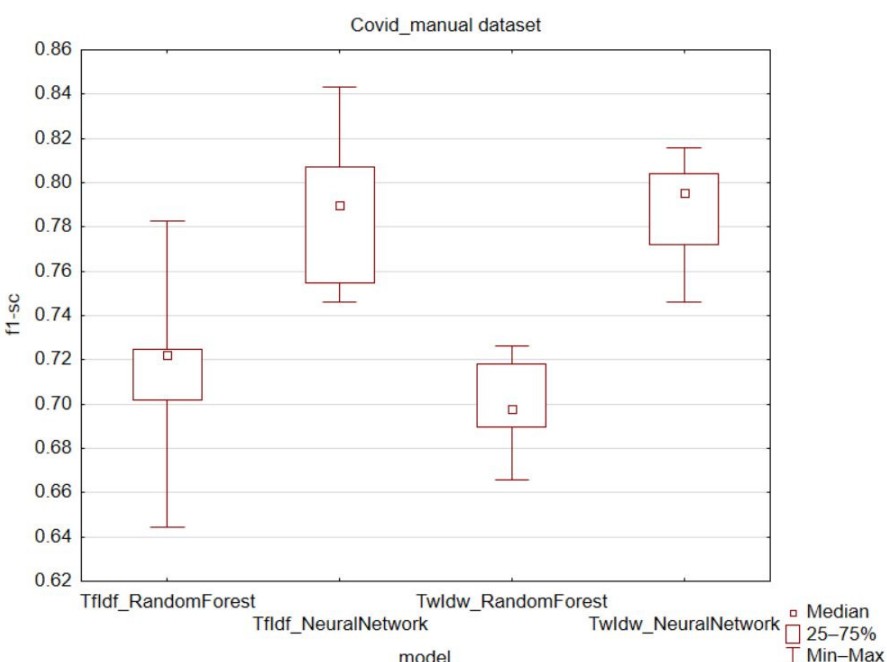

**Figure 10.** F1 score metric for the COVID_manual dataset.

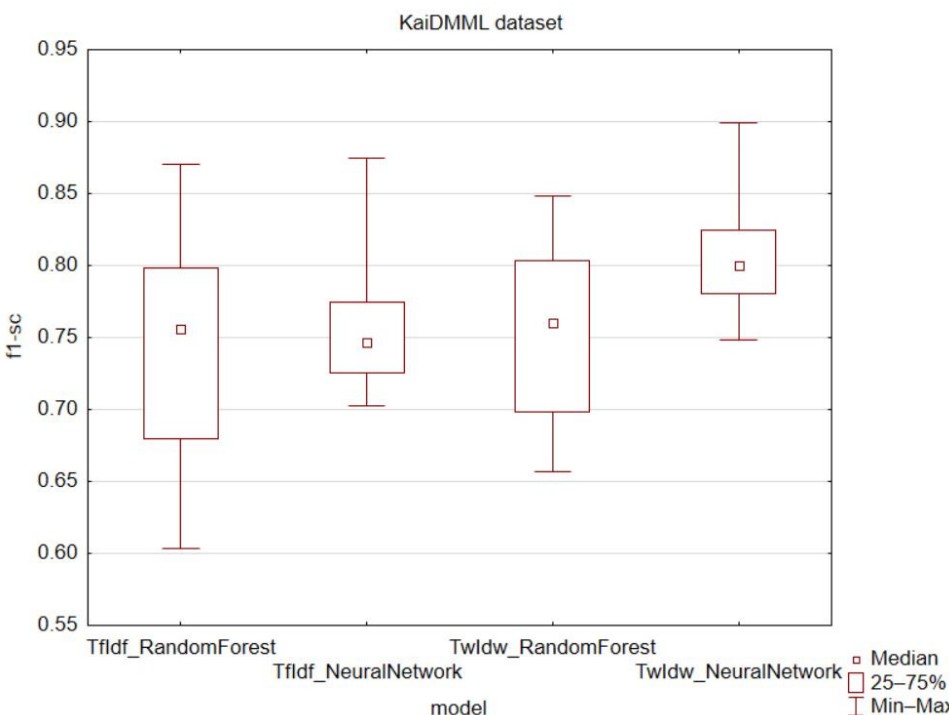

**Figure 11.** F1 score metric for KaiDMML dataset.

Better results in the case of the TwIdw method can be observed from the descriptive statistics. Due to this observation, the decision was made to assess the statistical significance of the differences between the TwIdw and TfIdf techniques. Since the metrics of precision and recall are encompassed in the f1 score metric, and for the sake of simplicity, only this performance measure was evaluated.

The Wilcoxon matched pairs test was used to test null statistical hypotheses claiming that the performance of models is independent of their representation [30,31]. The hypotheses were tested on results from all datasets as well as on results from individual datasets. The null hypothesis (H0) is not rejected (Tables 3–6), regardless of the

dataset (total/COVID_auto/COVID_manual/KaiDMML), or the model type (neural network/Random Forest).

**Table 3.** Wilcoxon matched pairs test in total.

| Total | Valid N | T | Z | *p*-Value |
|---|---|---|---|---|
| f1-sc(RF_TfIdf) and f1-sc(RF_TwIdw) | 30 | 230 | 0.051 | 0.9590 |
| f1-sc(NN_TfIdf) and f1-sc(NN_TwIdw) | 30 | 156 | 1.573 | 0.1156 |

**Table 4.** Wilcoxon matched pairs test for the COVID_auto dataset.

| COVID_auto | Valid N | T | Z | *p*-Value |
|---|---|---|---|---|
| f1-sc(RF_TfIdf) and f1-sc(RF_TwIdw) | 10 | 19 | 0.866 | 0.3863 |
| f1-sc(NN_TfIdf) and f1-sc(NN_TwIdw) | 10 | 27 | 0.051 | 0.9594 |

**Table 5.** Wilcoxon matched pairs test for the COVID_manual dataset.

| COVID_manual | Valid N | T | Z | *p*-Value |
|---|---|---|---|---|
| f1-sc(RF_TfIdf) and f1-sc(RF_TwIdw) | 10 | 18 | 0.968 | 0.3329 |
| f1-sc(NN_TfIdf) and f1-sc(NN_TwIdw) | 10 | 26 | 0.153 | 0.8785 |

**Table 6.** Wilcoxon matched pairs test for the KaiDMML dataset.

| KaiDMML | Valid N | T | Z | *p*-Value |
|---|---|---|---|---|
| f1-sc(RF_TfIdf) and f1-sc(RF_TwIdw) | 10 | 19 | 0.866 | 0.3863 |
| f1-sc(NN_TfIdf) and f1-sc(NN_TwIdw) | 10 | 9 | 1.886 | 0.0593 |

## 5. Discussion

In this experiment, the aim was to determine whether information obtained from syntactic analysis is a superior tool for constructing an input vector for a classifier, compared to information solely based on word frequency in texts. The calculation of the traditional TfIdf was modified into a suitable form that incorporates information about dependency grammar, and this technique was named TwIdw.

By employing this technique in the classification process, a reduction in the quartile and variation range of the resulting values was observed when compared to the reference technique TfIdf. The average accuracy reached similar values as in the case of the TfIdf technique ($\pm 1\%$), except for the feedforward neural network method in combination with the KaiDMML dataset, where an increase of up to 3.9% was observed. The random forest method in combination with TwIdw was less successful than the neural network method. Random forest in combination with the TwIdw technique achieved an increase only in the case of the KaiDMML dataset (1%). The feedforward neural network achieved an increase in classification accuracy using the TwIdw technique in all datasets.

Regarding the f1 score metric, the median value increased, and higher minimum and maximum resulting values were measured. The best classification accuracy values were achieved in the automatically evaluated COVID-19 dataset. The lowest accuracy values for the metric were achieved in the political dataset.

It is believed that this is attributed to the high variability of political topics present in the dataset. Better results can be achieved in a dataset that focuses on a specific area (in the case of the other datasets used, it is the virus COVID-19).

To present a comprehensive view of the findings, the precision and recall performance measures are presented in the results. The f1-score, derived from both precision and recall, is regarded as the most significant, thus considering the results for the f1-score as the most important is believed. However, precision and recall also confirmed good results for the technique TwIdw, especially in combination with neural network.

## 6. Conclusions

Based on the analysis of the related work, it is evident that fake news classification using NLP techniques is an active and challenging research area. Researchers have proposed various techniques such as feature-based methods, deep learning models, and hybrid approaches to tackle the problem. The effectiveness of these techniques is evaluated on different datasets, and the results show promising performance in detecting fake news.

In this research, a novel technique named TwIdw was proposed, which combines TfIdf and values of term depths extracted from the dependency grammar using the UDPipe tool. The proposed technique was evaluated on several datasets, and its performance was compared with existing techniques. The results demonstrate that TwIdw outperforms the technique TfIdf in terms of precision, recall, and F1 score.

The experimental analysis also highlights the impact of different parameters on the performance of the proposed technique. It was found that the performance of the classification models is significantly affected by the choice of algorithm, dataset, and feature selection techniques.

In conclusion, the proposed TwIdw technique is a promising approach for fake news classification using NLP. The results demonstrate the effectiveness of the proposed technique and its potential to be applied in real-world scenarios. There is a limited number of manually annotated unstructured texts; thus, further research needs to be considered using a larger quantity of such texts. Moreover, future work can explore further optimization of the proposed technique and evaluate its effectiveness in a wider range of applications.

**Author Contributions:** Conceptualization, K.S.N. and J.K.; methodology, K.S.N. and J.K.; validation, K.S.N. and J.K.; formal analysis, K.S.N. and J.K.; resources, K.S.N. and J.K.; data curation, K.S.N. and J.K.; writing—original draft preparation, K.S.N. and J.K.; writing—review and editing, K.S.N. and J.K.; visualization, K.S.N.; supervision, J.K.; project administration, J.K.; funding acquisition, J.K. All authors have read and agreed to the published version of the manuscript.

**Funding:** This work was supported by the Slovak Research and Development Agency under the contract no. APVV-18-0473.

**Institutional Review Board Statement:** Not applicable.

**Informed Consent Statement:** Not applicable.

**Data Availability Statement:** The datasets used in this study are freely available online at: https://towardsdatascience.com/explore-covid-19-infodemic-2d1ceaae2306 (accessed on 3 April 2023), https://data.mendeley.com/datasets/zwfdmp5syg/1 (accessed on 3 April 2023) and https://github.com/KaiDMML/FakeNewsNet (accessed on 4 April 2023).

**Conflicts of Interest:** The authors declare no conflict of interest.

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
