# Peer review of "TwIdw—A Novel Method for Feature Extraction from Unstructured Texts"

_applsci, doi:10.3390/app13116438_

Round 1

Reviewer 1 Report

The article presents improvements of standard methods in combination with syntactic approach in detection of fake news, especially NN proved to slightly improve detection. The is clear and concise, I would recommend to enrich abstract with clear indication of NN improvements (conclusion elaborates this, but abstract is important for future readers to have overall idea about the content).

The state-of-the art is solid, maybe authors should check the references for complete information, some literature is missing important bibliographic information, which may cause problems for readers to find the correct article, especially in recent content. The extent is sufficient, but I think the theme is now extensively studied, so authors could extend it, if possible.

The presented approach in experiments could also show some real-life examples of syntacic trees, not only model examples. The experiments are explained in standard way, it could also extend the descriptions of used methods in more illustrative way and compare its complexity, both theoretical and algorithmical.

Statistical analysis of results significance should be added, since the improvements are modest, it would be helpful for readers to assess authors' results also from this point of view (even the differences are not stat. significant).

Overall impression from the article is good, they use improved methods for detection, with some polishing of the article as descibed above (better explanation of methods, more thorough statistical analysis etc.), I can recommend the article for publication as an interesting comparison and slight improvements in important task.

The english is sufficient to undestand the content, I am not a native english speaker, but teh english is on standard technical level. Sentence sequence sometimes is not elegant (for me as a slavic language speaker it is not a problem), but maybe authors should ask a native speaker or ICT english expert to slightly corretc the article from stylistic point of view.

Author Response

Dear Reviewer, please see the attached pdf with all of our response.

Thank you

Reviewer 2 Report

The work looks interesting however lot of refinement need to make it in a proper standard. Please find below comments:

1)     Abstract need proper details such as existing model accuracy and proposed model accuracy etc.

2)     Introduction part has to be explained in a structured way, please follow good papers in nlp area and try to arranged accordingly.

3)     Literature survey requirement major work, need to refer more related article to justify your work. Few of the related  works can be refer in literature section:

a.      https://www.mdpi.com/2076-3417/12/16/8105

b.      https://www.sciencedirect.com/science/article/pii/S266709682200088X

c.      https://www.sciencedirect.com/science/article/pii/S2667096821000124?via%3Dihub

d.      https://link.springer.com/chapter/10.1007/978-981-16-7182-1_8

e.      https://www.techscience.com/cmc/v72n3/47476

4)     Avoid bullet points, write in paragraph. Methodology section need to be structured properly. Looks like project report.

5)     Datasets:  Don’t give the data source web link, better refer it from references, doesn’t looks nice.

6)     So TF-IDF and TW-IDW , what is the different between both , term and word both are same in NLP. How TW-IDW will be a better one than existing one, please justify this part properly.

7)     Why on F-1 Score, what about accuracy, precision, and recall, please provide all performance metrics.

8)     Discussion section is not containing any references, what researcher are discussing here, you have to justify the betterment of your work with existing one, so cite those papers and explain how your work is the best out of them. 

English must be improved. 

Author Response

(The authors gave the same response as above.)

Reviewer 3 Report

This manuscript proposed a novel feature extraction method based on TF-IDF for fake news detection. The experimental results show proposed TwIdw outperforms the traditional TF-IDF method. However, I found many main parts get missing in this manuscript. Here are my reasons why I decide to reject it.

1. The purpose of this manuscript in the introduction part is unclear. As I know, this manuscript aims to overcome the limitations in fake news detection tasks. However, the introduction about fake news is missing. According to the content described in the manuscript, it is more like aims to overcome the limitations of NLP tasks.

2. The limitations of previous studies are not clear. "However, there is still room for improvement, and future research should focus on developing more effective feature extraction techniques and exploring new deep learning architectures to further enhance the performance of these models." It seems like a general way of expressing it; the authors should describe the limitations in detail.

3. There is no continuity in the description of the related works part.

4. The proposed TwIdw method seems too simple, and why the authors proposed such a method is unclear. In other words, the reason why replacing term frequencies with the depth of the words is unclear.

5. The experiments are designed too simply. As mentioned, this manuscript aims to overcome the limitations of fake news detection. However, the benchmark is only the TF-IDF. I suggest the authors compare the performance with some methods related to fake news detection tasks.

I suggest the authors modify the format of the tables, formulas, and sentences with regulations.

Author Response

(The authors gave the same response as above.)

Round 2

Reviewer 2 Report

Authors have fulfilled the recommended comments  and accepted the publication. 

English is fine

Reviewer 3 Report

Although the authors had revised the manuscript under my suggestions in detail, I still decided to reject this manuscript. TF-IDF can be seen as the most typical method in NLP, which has been applied over decades. Although the authors conducted various experiments to prove the effectiveness of the proposed TwIdw, the proposed method is contrary to the concept of "novel." And if most previous studies detect fake news based on TF-IDF,  the proposed TwIdw seems appropriate. However, as the authors mentioned, it seems not. Overall, I don't think TwIdw is a "novel" method for detecting fake news, so I still reject it.

I suggest the authors revise the content. Some parts are hard to understand. And I suggest the authors modify the sentences with regulations.
